# Pre and Post Pandemic Depressive and Anxious Symptoms in Children and Adolescents in Northern Chile

**DOI:** 10.3390/jcm12041601

**Published:** 2023-02-17

**Authors:** Alejandra Caqueo-Urízar, Diego Atencio-Quevedo, Felipe Ponce-Correa, Patricio Mena-Chamorro, Alfonso Urzúa, Jerome Flores

**Affiliations:** 1Instituto de Alta Investigación, Universidad de Tarapacá, Antofagasta 1520, Arica 1000000, Chile; 2Escuela de Psicología y Filosofía, Universidad de Tarapacá, Arica 1000000, Chile; 3Departamento de Psicología, Universidad de la Frontera, Temuco 4811230, Chile; 4Escuela de Psicología, Universidad Católica del Norte, Antofagasta 1270460, Chile; 5Centro de Justicia Educacional (CJE), Pontificia Universidad Católica de Chile, Santiago 8940855, Chile

**Keywords:** COVID-19, mental health, secondary school

## Abstract

The psychological effects of the COVID-19 pandemic still represent a focus of concern, especially in children and adolescents who are a group particularly vulnerable to the psychological consequences of the COVID-19 pandemic, mainly due to the loss of socialization and leisure spaces. The aim of the study is to determine the variation in the levels of depressive and anxious symptomatology in children and adolescents in the North of Chile. Methods: A Repeated cross-sectional design (RCS) was used. The sample consisted of a total of 475 students aged 12 to 18 years (high school) from educational establishments in the city of Arica. To evaluate the changes in the mental health of students associated with the COVID-19 pandemic, the same mental health measures applied to students were compared in two waves (2018–2021). Results: An increase in the symptomatology levels of depression, anxiety, social anxiety, and problems with the family, while a decrease in problems with school and peers was observed. Conclusions: The results show that there is an increase in mental health problems associated with the periods of time in which the COVID-19 pandemic transformed the social relation spaces and classrooms of secondary school students. The observed changes point to future challenges, which include that it may be important to improve the coordination and integration of mental health professionals in educational centers and schools.

## 1. Introduction

The coronavirus disease pandemic (COVID-19) continues to spread with millions of people infected and dying from complications of the disease [1].

Existing research postulates that in addition to the physical health symptoms generated by the virus, there are increases in the prevalence of psychiatric and neuropsychiatric manifestations in those infected with COVID-19 and in the general population [2], including delirium, confusion, cognitive impairment, sleep disturbances, but especially symptoms of depression and anxiety [2,3].

A group especially vulnerable to the psychological consequences of the COVID-19 pandemic are children and adolescents, mainly due to the loss of socialization and leisure spaces [4,5] as well as to important changes in the lifestyle of this population, where physical activity time decreased, while screen time, caloric intake, and body weight tended to increase [6,7].

In this way, studies carried out in a child and adolescent population showed how the pandemic situation affected the mental health of the population. Thus, a study conducted in China during the initial stages of the pandemic, between February and March 2020, showed that there was already a higher prevalence of symptoms of depression and anxiety than before the COVID-19 pandemic [8]. In Italy, it was observed that in the second year since the beginning of the pandemic, anxious, depressive, and psychosomatic symptoms had increased, particularly in children and adolescents with unfavorable psychosocial conditions [9]. Furthermore, a study conducted with Scottish adolescents indicated that older, female adolescents who currently or previously received mental health support or additional support at school and adolescents who reported worse relationships at home since the COVID-19 pandemic were more likely to reach clinical threshold levels for anxiety, depression, and posttraumatic stress [10].

Among the possible causes are contextual problems, understood as those related to peers, school, or family members. In this sense, research has indicated that groups of students with a low parental education, restricted living conditions, migratory background, and parental mental health problems had a significantly higher risk of deterioration in health-related quality of life and mental health [11]. Moreover, evidence has been found that violence and maltreatment toward children and adolescents have increased during the pandemic [12,13].

A three-wave longitudinal study conducted among children and adolescents in Germany showed that the prevalence of low health-related quality of life and symptomatology of depression and anxiety increased significantly between the first and second waves, and trended downward in the third wave, when confinement measures were relaxed [14].

In Chile, mental health issues in the population are not rare. While the younger population has been considered as one of the least vulnerable at the level of physical symptoms and mortality rates, research has explored the impact of crisis and confinement in regard to study habits, academic performance and stress, social relationships and family ties [15]. In this sense, the results indicate that feelings of fear, worry, and anguish, and compulsive food intake and insomnia are symptoms that have increased considerably in the population over 14 years of age [16], while in children between 4 and 9 years of age, episodes of depressed mood, low energy, and changes in appetite have also increased [17]. Before the COVID-19 pandemic, only half of the Chilean mental health network had impatient facilities for critical child and adolescent psychiatry, 40% of them being located in the capital of the country (de la Barra et al., 2019). At the ambulatory level, the coverage represents about 28%, and different studies have stablished a prevalence of 22.6% for mental health disorders with impairment in children and adolescents from 4 to 18 years of age in Chile [18].

The objective of this study is to determine the variation in the levels of depressive and anxious symptomatology in children and adolescents in the North of Chile, in a pre- and post-pandemic context, in order to measure the effects of the health contingency on the mental health of this population and, thus, obtain valuable information for the creation of the necessary promotion, prevention, and intervention measures.

## 2. Materials and Methods

### 2.1. Participants

The sample consisted of 475 students aged 12 to 18 years old (secondary school) from educational establishments in the city of Arica. The participants were selected using a non-probabilistic convenience sampling strategy. Overall, the mean age was 14.3 (s.d = 1.7) years, 239 (52.5%) were female and 216 (47.5%) were male. Regarding the nationality of the participants, 404 (88.8%) were Chilean and 51 (11.2%) were foreigners. Likewise, 167 students reported belonging to an ethnic group (36.7%), of which 126 reported belonging to the Aymara ethnic group (27.7%).

The sociodemographic characteristics of the group of students surveyed before the pandemic in 2018 (*n* = 249) have a mean age of 14.42 years (s.d = 1.83), 50.2% identified themselves as male (*n* = 125) and 49.8% as female (*n* = 124). In total, 47.8% of the students presented a high vulnerability index (*n* = 119), 10% of the students were of foreign nationality (*n* = 25), 32.9% self-identified as belonging to an ethnic group (*n* = 82), and 70.3% of the students reported belonging to a religion (*n* = 175). The sociodemographic characteristics of the students surveyed in 2021 (*n* = 206) presented a mean age of 14.23 years old (s.d = 1.59), 55.3% identified themselves as male (*n* = 114) and 44.7% as female (*n* = 92). In total, 52.4% of the students had a high vulnerability index (*n* = 108), 12.6% of the students were of foreign nationality (*n* = 26), 41.3% self-identified as belonging to an ethnic group (*n* = 85), and 55.3% of the students reported belonging to a religion (*n* = 114). Table 1 details the sociodemographic characteristics.

### 2.2. Instruments

*Child and Adolescent Assessment System (SENA)* [19]. This is an instrument originally developed in the Spanish language. It has different versions according to the age of the students, as well as versions for different informants, including student self-report, family, and school. The instrument is composed of nine questionnaires, in order to assess a broad spectrum of emotional and behavioral problems across three age groups: infant (3–6 years), primary (6–12 years), and secondary (12–18 years). The response options for each version correspond to behavioral statements in a 5-point Likert format (1 = “Never” to 5 = “Always”). Recently, Sánchez-Sánchez et al. [20] have found that the reliability of SENA’s subscales is above 0.7 in Spain, making it suitable for use in children and adolescents in Spanish-speaking contexts. The internal consistency values for the study sample yielded acceptable reliability values for the total scale (α = 0.91). Similarly, acceptable reliability values were found for each of the SENA subscales: contextual problems with peers and family (α = 0.71), internalizing problems such as depression, anxiety, and social anxiety (α = 0.89), and externalizing problems (α = 0.82).

The instrument has a multidimensional approach. For the development of this research, the following dimension of the SENA were used in its secondary version (12–18 years). (A) Depression: measures the presence of depressive symptomatology, which is characterized by dysphoric mood, anhedonia, feelings of worthlessness, guilt, and thoughts associated with death. This scale contains items such as “I suffer a lot”. High scores suggest the presence of depressive manifestations, expressed by a sad or irritable mood. (B) Anxiety: evaluates the presence of generalized subjective discomfort characterized by persistent and recurrent worries typical of generalized anxiety through items such as “I am distressed or overwhelmed by my problems”. High scores suggest that students manifest feelings of nervousness, general subjective discomfort, and physiological overactivation. (C) Social anxiety: measures the presence of anxious symptoms specifically associated with social situations, where people fear being evaluated or judged. This scale contains items such as “I get nervous when there are a lot of people around”. High scores indicate that students show discomfort, nervousness, and insecurity in social settings.

The instrument was obtained from the country’s official distribution agency TEA Ediciones, and, therefore, the necessary permissions were obtained from the respective authors to use the scales at every moment.

*Ad-hoc sociodemographic scale:* Used to identify gender, age, nationality of students, ethnicity, and date of application.

### 2.3. Procedure and Design

A repeated cross-sectional design (RCS) was used [21]. A repeated cross-sectional design is a type of survey that uses data in which the same information is asked of an independent sample in each wave. In the case of an annual survey, this means that respondents in one year are different people from those in a previous or subsequent year. Therefore, there is little overlap in the sample between different periods. Samples in the RCS data can be collected consecutively or at irregular intervals over time. To assess changes in student mental health associated with the COVID-19 pandemic, data from 249 high school students aged 12–18 years who were administered SENA during the 2018 school year (Measure 1) and data from the administration of SENA to 206 high school students aged 12–18 years who were in their school year during 2021 (Measure 2) were used.

Incidental or convenience sampling was conducted. Then, 42 educational establishments in the city of Arica were invited to participate in the study. A total of 69% agreed to participate, considering 29 establishments in total. Once permission was granted to enter the regular meetings with the school directors, the parents were asked for their consent after the purpose and scope of the study were explained. Finally, the evaluation within each course group was scheduled with the directors. Before starting, the consent of the students themselves was requested among those who were already authorized to participate. Students answered in paper and pencil format. At least two trained interviewers were present to answer the doubts, together with the teacher of the same course. The duration was approximately 45 min.

### 2.4. Statistical Analysis

Initially, to characterize the sample, the proportions of each categorical variable were obtained and the mean, standard deviation, minimum and maximum, skewness, kurtosis, and Shapiro–Wilk normality test [22] were calculated for each continuous variable. Parametric comparative analyses were used, since the t-statistic is sufficiently robust under conditions of skewness and with large sample sizes (*n* > 60) [23,24]. A *t*-test for independent samples was used comparing the differences in the mean scores of the depression, anxiety, and social anxiety scales according to the year of application (proxy to measure the effect of the pandemic).

Finally, multiple linear regression models were run using a two-block hierarchical strategy. The first block included sociodemographic variables (religion, ethnicity, year of application, nationality, vulnerability index, Aymara) and another block with contextual variables (peers problems, family problems, and school problems). Three models were estimated, one for depression, anxiety, and social anxiety, respectively. Each model incorporated the standardized beta coefficients, which represent the changes in the standard deviation of the criterion variable. The predictor variables with the largest standardized beta coefficients suggest a larger relative effect on students’ depression, anxiety, and social anxiety. The presence of multicollinearity among the independent variables was ruled out by the tolerance level and the inflated variance factor (IVF), which was greater than 0.1 and less than 10 for all of them, respectively. The residuals were independent of each other. Homoscedasticity was confirmed by a scatter plot of the predictors and standardized residuals. Normality of the residuals for each dependent variable was tested by a histogram and Q-Q plot of the standardized residuals.

Statistical hypothesis testing of the data analyses was performed at a 5% significance level. All statistical analyses were performed using IBM SPSS version 25 software [25].

## 3. Results

The values of means, standard deviations, asymmetry, and kurtosis shown in Table 1. Symmetry and kurtosis were outside the acceptable ranges to be considered as normally distributed.

The *t*-test for independent samples showed statistically significant differences in all SENA scales used in the sample of secondary students (Table 2). A significant increase was observed in depression scores (2018: M = 2.11 [s.d = 0.86]; 2021: M = 2.47 [s.d = 1.01]; t(student) = −4.078; *p* = 0.000), similarly an increase was observed in anxiety (2018: M = 2. 48 [s.d = 0.84]; 2021: M = 2.92 [s.d = 1.00]; t(student) = −5.017; *p* = 0.000) and in social anxiety (2018: M = 2.38 [s.d = 0.77]; 2021: M = 2.79 [s.d = 0.98]; t(student) = −4.969; *p* = 0.000). The results reflect worsening mental health among students during the time period associated with the COVID-19 pandemic.

Table 3 shows that all linear regression models were significant. The linear regression model that controlled for the effects of sociodemographic variables (age, gender, year of application, vulnerability index, nationality, ethnicity, and Aymara) explained 9% of the variance of depression (F = 5.858, *p* = 0.000), 14% of the variance of anxiety (F = 10.560, *p* = 0.000), and 9% of the variance of social anxiety (F = 7.081, *p* = 0.000). Similarly, the model controlling for the effects of contextual variables (family problems, school problems, and peer problems) explained 54% of the variance in depression (F = 50.120, *p* = 0.000), 38% of the variance in anxiety (F = 27.050, *p* = 0.000), and 21% of the variance in social anxiety (F = 12.250, *p* = 0.000).

Table 4 details the standardized coefficients. The results of the multiple regression showed that in all models the year of application had a statistically significant effect. Therefore, when controlling sociodemographic and contextual variables, it could be observed that the effect of the pandemic increased depressive symptomatology, anxiety, and social anxiety.

On the other hand, on Table 5, the analysis of standardized coefficients showed that females (β = −0.238; *p* = 0.000), family problems (β = 0.432; *p* = 0.000), school problems (β = 0.325; *p* = 0.000), and problems with peers (β = 0.140; *p* = 0.000) increased depression in students. Moreover, the results showed that female gender (β = −0.288; *p* = 0.000), lower vulnerability (β = 0.044; *p* = 0.044), family problems (β = 0.269; *p* = 0.000), school problems (β = 0.219; *p* = 0.000), and peer problems (β = 0.176; *p* = 0.000) increased anxiety in students. Finally, standardized coefficients yielded that female students (β = −0.231; *p* = 0.000), family problems (β = 0.163; *p* = 0.000), school problems (β = 0.136; *p* = 0.000), and peer problems (β = 0.170; *p* = 0.001) increased social anxiety in students.

## 4. Discussion

The aim of this study was to determine the variation in the levels of depressive and anxious symptomatology in children and adolescents from Northern Chile, in a pre- and post-COVID-19 pandemic context. The results indicate an increase in the symptomatology levels of depression, anxiety, social anxiety, and problems with the family, while a decrease in problems with school and peers was observed.

Regarding depressive symptomatology, the increase in levels prior to the pandemic context was congruent with longitudinal studies conducted in Germany [2] and Italy [9].

These results seem to be more significant when subjects come from vulnerable socioeconomic strata, as has been observed in other regions of the world, as a risk factor both prior to the pandemic, but even more so after its outbreak, when social and economic gaps became more noticeable [11,26,27].

Regarding anxiety symptoms, their increase was also in line with research on the subject, particularly with the meta-analysis carried out by Samji et al. [28]. The literature points to various causes for the increase in anxious symptomatology, highlighting the fear of infection of oneself or a loved one, the recent implementation of containment measures and concerns related to the economic and social repercussions of the pandemic [14,29,30].

On the other hand, a significant finding of the study had to do with the increase in the levels of social anxiety in the pandemic context. This event was observed and explained by Zheng et al. [31] based on the Stimulus–Organism–Response (SOR) model, postulating that, in a confined environment, people exhibit more anxiety, and that this confinement context strengthens the negative effects of the severity of the pandemic, increasing social anxiety due to the possibilities of contagion that interactions with other people entail [31].

The increase in family problems has been widely observed in the literature [28]. The COVID-19 pandemic has meant challenges not only for children and adolescents, but also for their parents and close environment. In this sense, concern about possible infections, death of loved ones, job instability, and parental difficulties have generated a negative impact on family life, while these coexistence problems have emerged as mediating variables between the pandemic and the deterioration in mental health indicators [32,33,34].

The results of this study indicate a decrease in the frequency of problems with school and classmates, which may have its origin in the decrease of academic demands within the curricular adjustments made by the Ministry of Education and the educational establishments during the pandemic, as well as in the decrease of interactions with classmates, which, therefore, meant less conflicts with them. However, these variables should be studied, because in the return to face-to-face classes, significant increases have been observed regarding problems in students’ relationships with educational establishments and their classmates.

The implication of this study was to obtain longitudinal information on mental health indicators in a sample of children and adolescents from northern Chile in the context of the COVID-19 pandemic. These results are an important initial approximation for knowing the trajectories of these indicators, and in this way creating adequate intervention strategies, as well as promotion and prevention strategies focused on mental health.

While the COVID-19 pandemic continues to unfold, there is evidence of psychological interventions that have been effective in other epidemiological outbreaks with child and adolescent populations, including art therapy, play therapy, yoga-based therapy, and collaboration with child psychology experts [35]. The limitations of the study were, on the one hand, the small sample size, which reduces the generalizability of the results obtained; secondly, the fact that the instruments applied are self-reporting, so that external sources of information such as the parents or teachers of the participating students are not taken into account; and thirdly, that the study only considered two rounds of data, while the COVID-19 crisis is a phenomenon that have been experienced in several stages of changes during the last years. Finally, in this study, the effect attributable to the COVID-19 pandemic was measured using year of application as a variable to be introduced in regression line; although the results reflect a relevant effect on dependent variables, longitudinal measures differ greatly from the repeated cross-sectional design used in this study, in which an independent sample is collected in each wave to represent the population during that time period. Thus, results should be interpreted with caution. Future research should be carried out with representative sample sizes, taking into account multiple sources of information and exploring the impact of the pandemic in its different stages in various round surveys rather than one round only.

## 5. Conclusions

The psychological effects of the COVID-19 pandemic still represent a focus of concern, especially as regards children and adolescents who are a group that is particularly vulnerable to the psychological consequences of the COVID-19 pandemic, mainly due to the loss of socialization and leisure spaces. The changes observed in the trajectories of mental health indicators in the groups of students before and after the COVID-19 pandemic indicated an increase in the levels of depressive symptomatology, anxiety, social distress, and problems with family, while a decrease was observed in problems with school and peers. The results show that there is an increase in mental health problems associated with the periods of time in which the COVID-19 pandemic transformed the social relations spaces and classrooms of secondary school students. The mental health of children and adolescents was already an issue of concern before the outbreak of the pandemic, and the crisis generated by the virus aggravated a pre-existing problem. The observed changes point to future challenges, which include that it may be important to improve the coordination and integration of mental health professionals in educational centers and schools, but also, with public politics, which improves coverage of mental health services for children and the adolescent population.

## Figures and Tables

**Table 1 jcm-12-01601-t001:** Sociodemographic characteristics of the sample.

		2018 (*n* = 249)	2021 (*n* = 206)
Age		14.42 (s.d = 1.83)	14.23 (s.d = 1.59)
Gender			
	Men	125 (50.2%)	114 (55.3%)
	Women	124 (49.8%)	92 (44.7%)
Vulnerability index			
	Low	130 (52.2%)	98 (47.6%)
	High	119 (47.8%)	108 (52.4%)
Nationality			
	Chilean	224 (90%)	180 (87.4%)
	Foreign	25 (10%)	26 (12.6%)
Ethnicity			
	Yes	82 (32.9%)	85 (41.3%)
	No	167 (67.1%)	121 (58.7%)
Aymara			
	Yes	58 (23.3%)	68 (33%)
	No	191 (76.7%)	138 (67%)
Religion			
	Yes	175 (70.3%)	114 (55.3%)
	No	74 (29.7%)	90 (43.7%)

**Table 2 jcm-12-01601-t002:** Descriptive statistics.

	M (SD)	Min-Max	S	K	Shapiro–Wilk
Age	14.3 (1.72)	12–18	0.338	−0.975	**0.921 ***
Depression	2.27 (0.95)	1–5	0.818	−0.124	**0.925 ***
Anxiety	2.68 (0.94)	1–5	0.558	−0.431	**0.960 ***
Social anxiety	2.57 (0.89)	1–5	0.586	−0.161	**0.962 ***
Family problems	1.80 (0.72)	1–5	1.420	2.073	**0.869 ***
School problems	1.79 (0.68)	1–5	1.432	2.312	**0.873 ***
Peer problems	1.29 (0.43)	1–5	2.605	8.212	**0.676 ***

Note: M = Mean; SD = Standard deviation; Min-Max = Minimum and maximum; S = Standardized skewness; K = Standardized kurtosis; Shapiro–Wilk = Shapiro–Wilk test; Values in bold indicate a statistically significant (*p* < 0.05); Values in bold and asterisk (*) indicate a statistically significant (*p* < 0.001).

**Table 3 jcm-12-01601-t003:** Means differences according to the year of evaluation (2018–2021).

	2018	2021	t	df	Sig
	Mean (s.d)	Mean (s.d)			
Depression	2.11 (0.86)	2.47 (1.01)	−4.078	453	0.000
Anxiety	2.48 (0.84)	2.92 (1.00)	−5.017	453	0.000
Social anxiety	2.38 (0.77)	2.79 (0.98)	−4.969	453	0.000

**Table 4 jcm-12-01601-t004:** Multiple linear regression.

	Model 1				Model 2			
	R	Adj. R^2^	df	F	R	Adj. R^2^	df	F
Depression	0.30	0.09	452	**5.858 ***	0.74	0.54	452	**50.12 ***
Anxiety	0.40	0.14	452	**10.56 ***	0.63	0.38	452	**27.05 ***
Social anxiety	0.33	0.09	452	**7.081 ***	0.48	0.21	452	**12.25 ***

Note: F = Statistical F; *p* = Significance; Adj. R^2^ = Coefficient R squared corrected; β = Standardized regression coefficient; Values in bold indicate a statistically significant (*p* < 0.05); Values in bold and asterisk (*) indicate a statistically significant (*p* < 0.001).

**Table 5 jcm-12-01601-t005:** Standardized coefficient.

	Depression		Anxiety		Social Anxiety		Collinearity
**Model 1**	**β**	** *p* **	**β**	** *p* **	**Β**	** *p* **	** *IVF* **
Age	−0.029	0.523	0.071	0.105	−0.012	0.787	1.011
Gender	−0.238	0.000	−0.288	0.000	−0.231	0.000	1.013
Year of application	0.175	0.000	0.230	0.000	0.201	0.000	1.041
Vulnerability index	0.034	0.478	−0.094	0.044	−0.002	0.972	1.147
Nationality	−0.023	0.625	−0.026	0.570	0.033	0.476	1.066
Ethnicity	0.005	0.947	0.056	0.455	−0.003	0.971	2.943
Aymara	0.048	0.538	−0.003	0.966	−0.076	0.329	3.018
Religion	0.028	0.554	0.005	0.918	0.037	0.431	1.089
**Model 2**	**β**	** *p* **	**β**	** *p* **	**Β**	** *p* **	** *IVF* **
Age	0.30	0.351	0.121	0.001	0.026	0.536	1.029
Gender	−0.233	0.000	−0.299	0.000	−0.247	0.000	1.104
Year of application	0.203	0.000	0.260	0.000	0.227	0.000	1.079
Vulnerability index	−0.007	0.831	−0.124	0.002	−0.023	0.609	1.178
Nationality	−0.006	0.853	−0.012	0.766	0.042	0.336	1.118
Ethnicity	−0.019	0.734	0.036	0.567	−0.018	0.803	2.946
Aymara	0.049	0.375	−0.001	0.982	−0.074	0.311	3.020
Religion	−0.047	0.159	−0.047	0.230	0.003	0.954	1.114
Family problems	0.432	0.000	0.269	0.000	0.163	0.001	1.294
School problems	0.325	0.000	0.219	0.000	0.136	0.008	1.510
Peer problems	0.140	0.000	0.176	0.000	0.170	0.001	1.427

Note: β = Standardized regression coefficient. *IVF =* Inflated variance factor.

## Data Availability

Not applicable.

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
