# Peer review of "Pre and Post Pandemic Depressive and Anxious Symptoms in Children and Adolescents in Northern Chile"

_jcm, 2023, doi:10.3390/jcm12041601_

Round 1

Reviewer 1 Report

Thank you for the opportunity to review the manuscript.

The rationale for this study has been provided, however, more information should be given about mental health of adolescents In Chile.

Instruments - it is not clear what tool have been used, Authors write "The set of questionnaires has a multidimensional approach, for the development of 100 this research the following questionnaires were used in their Secondary version (12-18 101 years)." No information about the psychometric properties of this scales has been provided.

There is no information in the manuscript about the detailed two groups demographic characteristics.

It is not clear if the regression models were calculated for the whole group? The Authors should explain and justify this calculation. If this is for the whole group, why the calculation was not made fot particular group? (the groups are different)

Discussion section should be broaden, especially the results of an appropriate regression analyses should be discussed in more details.

More practical conclusions should be proposed.

Reviewer 2 Report

This paper employs repeated cross-sectional design (RCS) to study the variation of psychological status due to the consequences of the COVID-19 pandemic among children and adolescents. The study will be valuable to study the social impact of the pandemic on the general population especially the vulnerable groups in different social context. The research is well introduced and the result of the surveys are clearly presented. I expect authors could address my following concerns in the revision. 

1.  The survey instruments are self-reporting. The pre-Covid results were collected during Covid period. How to decrease the effect of post-crisis(covid) memory on responding to the pre-covid survey? Any measures to decrease such psychological implication when collecting the first-round data?

2.  Though the authors reported the result of multicollinearity issue in Line155-157. It will be good to see the correlation tables of different independent variables in the two rounds of surveys. 

3.  In Discussion section, the authors referred to Zheng et al.[30] and explained the social isolation ie the confined environment or lockdown had generally negative effects on psychology status of the general population. While, the public had experienced several stages of changes during the pandemic in last three years. The covid control measures varied in different stages and different societies/countries. It would be good explore such impacts along the stages of the pandemic by some more rounds (during-pandemic) surveys rather than one round only. More interesting findings could be explored from the surveys.   But that will change the research design of the project. 

Reviewer 3 Report

Thank you for allowing me to review this interesting study. Yet, the following are some minor comments to reconsider before potential publication :

1. The sample size calculation needs to be clarified more

2. The psychometric properties of the measurement tool are missing 

3. Did you take permission to reuse the tool? 

Regards, 

Round 2

Reviewer 1 Report

The manuscript has been improved. I suggest to add the reliability of measurement tools in the current study (Cronbach's alpha).
